



# Soil $CO_2$ efflux errors are lognormally distributed - Implications and guidance.

Thomas Wutzler[1], Oscar Perez-Priego[1], Kendalynn Morris[1], Tarek El-Madany[1], and Mirco Migliavacca[1]

[1]Max Planck Institute for Biogeochemistry, Hans-Knöll-Straße 10, 07745 Jena, Germany

**Correspondence:** Thomas Wutzler
(twutz@bgc-jena.mpg.de)

**Abstract.**

Soil $CO_2$ efflux is the second largest carbon flux in terrestrial ecosystems. Its feedback to climate determines model predictions of the land carbon sink, which is crucial to understanding the future of the earth system. For understanding and quantification, however, observations by the most widely applied chamber measurement method need to be aggregated to larger temporal and spatial scales. The aggregation is hampered by random error that is characterized by occasionally large fluxes and variance heterogeneity that is not properly accounted for under the typical assumption of normally distributed fluxes. Therefore, we explored the effect of different distributional assumptions on the aggregated fluxes. We tested the alternative assumption of log-normally distributed random error in observed fluxes by aggregating one year of data of four neighbouring automatic chambers at a Mediterranean savanna-type site.

With the lognormal assumption, problems with error structure diminished and more reasonable confidence intervals were obtained. While the differences between distributional assumptions diminished when aggregating data of single chambers to an annual value, differences were important at short time scales and were especially pronounced when aggregating across chambers to plot level.

Hence we recommend as a good practice that researchers report plot-level fluxes with uncertainties based on the log-normal assumption. Model-data integration studies should compare predictions and observations of soil $CO_2$ efflux at log scale. This study provides methodology and guidance that will improve the analysis of soil $CO_2$ efflux observations and hence improve understanding of soil carbon cycling and climate feedbacks.

## 1 Introduction

Instantaneous measurements of soil $CO_2$ efflux, such as those made with automated respiration chambers, have gained importance for understanding ecosystem carbon dynamics in recent years (Phillips et al., 2016). Poor understanding of the feedbacks of this flux to global change introduces large uncertainties in the predicted terrestrial carbon sink and the projection of the earth system (Friedlingstein et al., 2014). Hence, observations and associated uncertainty estimates at the ecosystem scale have the potential to better resolve model structural uncertainty and predictive ability (Vargas et al., 2010). Among measurement device





enclosure types and configuration, chambers represent the most widely used approach for measuring pedon-scale soil $CO_2$ efflux (Livingston et al., 2006).

Derivation of ecosystem-scale $CO_2$ efflux, however, involves summarizing data of several chambers across space and several measurements across time, i.e. spatial and temporal aggregation. This aggregation poses problems in data analysis. Flux mea-

surements from several chambers, which are typically representative of an area below $1\mathrm{m}^2$, need to be aggregated to the plot level of hectares in order to compare them with ecosystem respiration inferred from eddy covariance based net land-atmosphere carbon fluxes (NEE) (Laville et al., 1999; Christensen et al., 1996; Held et al., 1990; Reth et al., 2005). Problems are indicated by the widespread finding of higher values for aggregated soil $CO_2$ efflux than NEE (Barba et al., 2018). Theoretically, up-scaled soil respiration should always be smaller than ecosystem respiration and NEE, because soil respiration is only a part of

ecosystem respiration, and NEE is always smaller than or equal to ecosystem respiration.

One challenge is spatial heterogeneity together with limited number of measurement locations, that together constrain the precision of the plot-level aggregated flux (Rodeghiero and Cescatti, 2008). Stronger spatial correlations or stronger correlations of soil $CO_2$ efflux with other more easily measurable spatially distributed variables could help with up-scaling. However, the differences between chambers a few meters apart can be as great as that between distant chambers (Giasson et al., 2013),

and correlation with soil moisture or plant activity is not sufficiently strong and changes across seasons (Leon et al., 2014; Fóti et al., 2016).

A second challenge is posed by a large component of random error. It originates from intrinsic fine-scale processes such as microbial metabolic pathways, gas diffusion, or microbial population dynamics and, to a smaller extent, from instrumentation (Lavoie et al., 2015; Pérez-Priego et al., 2015) (see section 2.2 for terminology). Random error is usually assumed to be

normally distributed, however, violation of this assumption poses problems for analysis and aggregation across space and time. A first problem is the increasing variance with increasing flux, which violates the assumption of homoscedasticity of variance, which is the base of many statistical tests. A second problem is the occurrence of strong tails, i.e., higher probability of large absolute errors (Savage et al., 2008; Cueva et al., 2015; Lavoie et al., 2015) compared to the normal assumption. This is often associated with hot spots and hot moments, i.e., locations or times where large fluxes occur at a small scale (Leon

et al., 2014; Vargas et al., 2018). To overcome these problems, Savage et al. (2008) proposed using the Laplace distribution. If this proposal is applicable depends on how the data will be used. For instance, model-data integration studies can use the Laplace assumption by using a cost-function that is based on the median absolute deviation rather than the squared difference (Richardson et al., 2006). However, other statistical methods still rely on the normal assumption. For instance, using mixed effects models (Pinheiro and Bates, 2000; Zuur et al., 2009) in aggregating measurements across several chambers requires the

normal assumption for a random effect to represent grouping in the data.

In this study, we tackle this challenge of analysing and aggregating flux data associated with random error. We evaluate the assumption of random error being lognormally distributed as an alternative to the assumption of additive random error from a normal or Laplace distribution.

The lognormal distribution describes measurements with a more or less skewed distribution. It is defined as a continuous

probability distribution of a random variable whose logarithm is normally distributed. Such distributions often arise when





values cannot be negative, such as with soil $CO_2$ efflux that is mainly driven by autotrophic and heterotrophic respiration. While the combination of complex additive processes or the sum of random numbers lead to normally distributed observations, a combination of multiplicative processes or the product of random numbers lead to lognormally observations (Limpert et al., 2001) . With the lognormal assumption, log-transforming observations allows further analysis using the normal assumption.

The objectives of this study are, first, to demonstrate that using the lognormal assumption leads to improved analysis, and second, to help readers to apply the lognormal assumption to their own data.

    Using observed fluxes of four automated soil $CO_2$ efflux chambers of a Mediterranean tree-grass savanna ecosystem, we compare the results of the lognormal approach to two traditional assumptions of normally or Laplace distributed random error. We show that the lognormal approach diminishes several problems: The lognormal approach leads to more reasonable

confidence bounds of aggregated fluxes while keeping continuity of expected values with previously published aggregated fluxes. Finally we discuss assumptions and the implications of our findings.

## 2   Methods

### 2.1   Study site and measurement

Data were collected at an established FLUXNET site (ES-LMa and ES-Lm1) near Majadas de Tiétar, Extremedura, Spain

(39°56'25.12" N, 5°46'28.70"W). In May of 2015, 16 semi-automated, soil efflux measurement chambers were installed in a stratified-random sampling design (Rodeghiero and Cescatti, 2008; Giasson et al., 2013; Phillips et al., 2016) grouped into different treatments and canopy positions. The chambers are a house-developed, stainless-steel design, connected to Li-Cor 820 (Li-Cor, Lincoln, Nebraska, USA) measuring on a half-hourly cycle. During this cycle one chamber at a time would close for a three-minute measurement duration. While there were 16 chambers in all, only data from four chambers in the open grassland

stratum within the control plot are used for the purposes of this manuscript. The aggregate across these four chambers, here, is referred to as plot-level estimate, although it only represents the open grassland. Flux rates were calculated from $CO_2$ concentration time series using the RespChamberProc R package[1] (correcting for water vapour dilution and estimating the initial slope of concentration increase). Data used in this manuscript range from November 2015 to November 2016. Additional details about the site can be found in El-Madany et al. (2018).

### 2.2   Distributional assumptions

Each measurement has uncertainty, and this uncertainty can be characterized by a density distribution. For similar environmental conditions, observed fluxes ($R_S$) scatter around a basic flux ($R_B$). The noise originates from both, instrumentation error (IE) and process error (PR), a stochastic component intrinsic to the measured soil system. While the nonsystematic component of IE is usually well described as a normally distributed random variable, PR can be described by a normal or laplace distribution

---

[1]https://github.com/bgctw/RespChamberProc





(1):

$$R_S = R_B + \epsilon_{PR,add} + \epsilon_{IE} \tag{1a}$$

$$\epsilon_{PR,add} \sim norm(0, \sigma_{PR,add}) \text{ or } \sim laplace(0, b) \tag{1b}$$

$$\epsilon_{IE} \sim norm(0, \sigma_{IE}), \tag{1c}$$

or alternatively with the lognormal distribution (2):

$$R_S = R_B \, \epsilon_{PR,mult} + \epsilon_{IE} \tag{2a}$$

$$\epsilon_{PR,mult} \sim lognorm(-\sigma^2/2, \sigma) \tag{2b}$$

$$R = log(R_S) \approx log(R_B) + log(\epsilon_{PR,mult}) \tag{2c}$$

where $\epsilon$ are error terms and $\sigma_{PR,add}$, $\sigma$, and $b$ are shape parameters of their respective distributions. $\epsilon_{PR,mult}$ is assumed to

be lognormally distributed with expected value of one. $\epsilon_{IE}$ is usually small compared to $R_B \epsilon_{PR,mult}$ (Lavoie et al., 2015) and
hence, approximation (2c) allows analysis of log-transformed observations.

## 2.3  Estimating random error

It is the distribution of the error terms, which is relevant for estimating uncertainty of aggregated fluxes. It should not be
confused with the distribution of measurement values. Error terms are the difference between observed fluxes and a true basic

flux. The true flux is unknown but can be estimated by the average flux at similar environmental conditions.

A simple method to estimate the error terms is daily differencing, excluding days with and after rain events (Savage et al.,
2008). The daily differencing method assumes that records 24 hours apart represent similar environmental conditions and hence
differences in the observed flux can be used to estimate the random error. It includes both, the non-systematic component of
instrumentation error, IE, and process error, PR (section 2.2).

An alternative method is the Lookup-Table approach (LUT). It is commonly used in the marginal distribution sampling
method (Reichstein et al., 2005), a method used for filling gaps in data from eddy covariance sensors (Wutzler et al., 2018).
When applied to soil $CO_2$ efflux observations in this study, similar environmental conditions were determined by the hour of
the day ($\pm 1$ hour), temperature ($\pm 3°C$), soil moisture ($\pm 5\%$), and a time window. The time window size was increased from
$\pm 1$ day to $\pm 3, 12, 24$ days until there were at least 5 valid measurements to average across.

A third alternative is modeling the base flux by its relationship with ancillary observations, such as temperature. We tried
modeling the $CO_2$ efflux temperature relationship with varying basal respiration (Gomez-Casanovas et al., 2013; Reichstein
et al., 2005). However, cross validation showed that this approach did not achieve good results for $R_S$ at the Majadas site,
because correlation with temperature is generally weak at water-limited sites (Vargas et al., 2018; Rey et al., 2011). Moreover,
during dry periods small precipitation events caused respiration pulses without observed concurrent increases in soil moisture

at 5 cm soil depth, where our measurement sensor was located.





When using the lognormal assumption, daily differencing was applied to the log-transformed observed fluxes. While for the LUT approach, the difference between observed and mean flux was computed with the log-transformed values $log(\epsilon_{PR,mult}) = R - log(R_{S,\mathrm{LUT}})$.

### 2.4  Estimating correlations in random error

The aggregation across time must take into account correlations among individual observations, because subsequent measurements are usually autocorrelated.

After computing the error terms, we computed the empirical autocorrelation function from the time series of error terms using the `acf` function implemented in `R` (Venables and Ripley, 2002). Only the first components of the autocorrelation function can be estimated reliably from the time series. Hence, we only used those components before the first negative autocorrelation

(Zięba and Ramza, 2011), to construct the variance-covariance matrix with components $\rho_{ij}$.

### 2.5  Gapfilling

Gaps in the flux time series have to be filled before computing the annual aggregated flux. Shorter gaps were filled using the LUT with a window size up to $\pm 24\ \mathrm{days}$ (section 2.4). Longer gaps were filled by fitting a random forest machine learning model (Vargas et al., 2018) with predictors "half hour of the day", global radiation, air temperature, soil temperature, precipita-

tion, vapor pressure deficit (VPD), mean daily soil temperature, mean daily air temperature, soil moisture, mean soil moisture for all chambers, and daylength. Gapfilling extrapolated at maximum 5 days into gaps. The remaining long gaps were treated as missing for spatial aggregation or replaced by the mean flux of the other chambers for temporal aggregation.

### 2.6  Aggregating fluxes with the normal assumption

If instrumentation error is dominating, the error is usually well described by a normal distribution with a mean of zero (1c).

Hence, when aggregating several independent measurements over time, the sum and the mean are normally distributed with expected values being the sum of the respective terms, and variance being computed by adding variances of measurements.

However, for time series usually one must consider autocorrelation, where successive measurements are not independent of each other, i.e where knowing one measurement holds information for predicting other measurements close in time. One has to subtract covariances when summing variances. For autocorrelated series this leads to formulas dependent on the effective

number of observations (5) based on the autocorrelation function, which describes how strongly measurements are correlated across time lags (Bayley and Hammersley, 1946; Zięba and Ramza, 2011).



$$\mathrm{Var}(\bar{x}) = \frac{\mathrm{Var}(x)}{n_{eff}} \tag{3}$$

$$\mathrm{Var}(x) = \frac{n_{eff}}{n(n_{eff}-1)} \sum_{i=1}^{n} (x_i - \bar{x})^2 \tag{4}$$

$$n_{eff} = \frac{n}{1 + 2\sum_{k=1}^{n-1} \left(1 - \frac{k}{n}\right) \rho_k}, \tag{5}$$

where $\bar{x}$ denotes the mean of a vector of random variable $x$, in our case soil respiration, $R_S$, $n$ is the number of records, and

$\rho_k$ denote the coefficients of the autocorrelation function. The autocorrelation function is usually not known, but their first components can be reliably estimated from the data. We followed Zięba and Ramza (2011) who recommend using only the components before the first negative component for $k$ in (5) instead of all $n-1$ components (section 2.4).

Confidence intervals of the probability for the aggregated mean flux were computed as $\bar{x} \pm 1.96\,\mathrm{sd}(\bar{x})$, where $\mathrm{sd}$ denotes the standard deviation.

## 2.7    Aggregating fluxes with the lognormal assumption

An overview of the properties of the lognormal distribution is provided in appendix A.

For spatially aggregating fluxes, we first log-transformed each observed flux, $R = \ln R_S$. Next, we used the log-transformed values of the same time from different chambers to compute the parameters $\mu$, and $\sigma$ of the spatial distribution (A4). Next, we used the distribution parameters to obtain a spatially aggregated estimate by computing the expected value (A2a) and

confidence bounds between quantiles 2.5% and 97.5% (A6).

For aggregating fluxes of a single chamber over time, we estimated parameters of lognormal distribution ($\mu_i$ and $\sigma_i$) for each half-hour, $i$, measurement from their moments (mean and variance across chambers) by (A5). Next, we used these parameters to compute the distribution parameters of the sum of fluxes (A7) using the R function `estimateSumLognormalSample` that we provide with the lognorm R package. The function takes arguments $\mu_i$ and $\sigma_i$, a vector of model errors to estimate the

correlations, and a vector that tells which of the terms are original measurements rather than gapfilled values. It computes the sum's standard deviation, $\sigma$, based on original measurements only, while computing the expected value and scale parameter, $\mu$, including gapfilled values. Hence, the expected value of the sum corresponded to the sum of the gapfilled measured fluxes (A7a).

## 3    Results

### 3.1    Distribution and scaling of random errors

The distribution of error terms obtained by daily differencing had strong tails, while when applying the daily differencing to log transformed values, $R = \ln R_S$, the distribution of the resulting error became closer to normal than at original scale (Fig. 1). For large negative outliers the lognormal distribution approximated error distribution even better than the Laplace

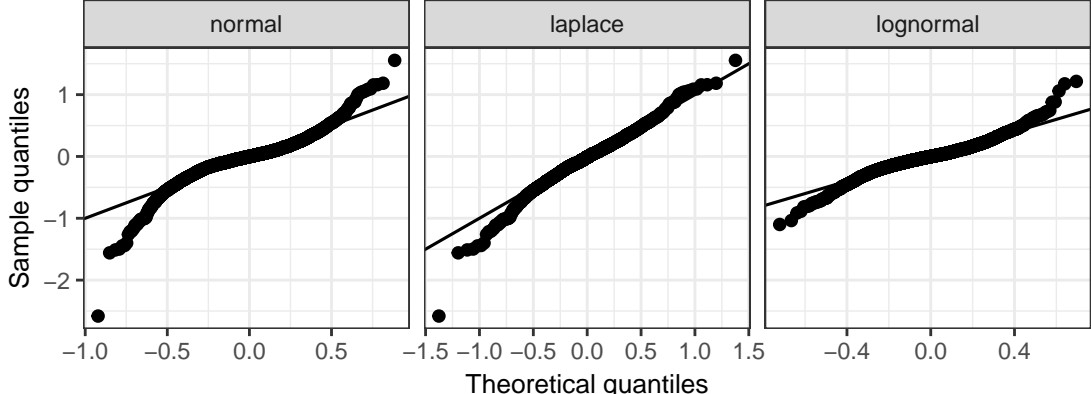

**Figure 1.** Quantile-Quantile plots compare the sample quantiles of observation error to theoretical distribution quantiles. The closer the points to the displayed 1:1 line, the better the approximation. Both the Laplace distribution with observations at original scale and the normal distribution with log-transformed observations (lognormal assumption) approximated the sample quantiles better than the normal assumption with observations at original scale.

distribution. Moreover, the log-transformation avoided the problematic scaling of random error with flux magnitude. Standard deviation across random error within one day scaled with flux magnitude at the original scale (Fig. 2 top) but did not scale at the log-transformed scale (Fig. 2 bottom).

## 3.2 Spatial aggregation

5 We compared the aggregation of half-hourly fluxes across four neighbouring chambers using the lognormal assumption (Section 2.7) versus using the normal assumption (Section 2.6). For periods without extreme fluxes, both the aggregated value and confidence intervals did not differ much unless the following features were present: first, the lower bound of the confidence interval was usually close to the lowest observed value, and second, the upper confidence bound was not as strongly influenced by high fluxes. Differences became more evident with high fluxes after rainfall when there was larger variability across cham-

10 bers. In such periods, the lognormal-based lower confidence bound was closer to the low measurements and avoided negative fluxes (Fig. 3).

## 3.3 Temporal aggregation of single chamber fluxes

The expected value of the aggregated fluxes across the 48 half-hourly measurements per day was the same across distributional assumptions. It corresponded to the mean of the observed values. The width of the 95% confidence intervals was the same for

15 most records but differed in few cases (Fig. 4, top panel).

Instances where the lognormal assumption resulted in much wider confidence intervals occurred on days with very low fluxes. In these cases the process error, which scales with the flux, is small compared to the instrumentation error and the assumption that error is dominated by the multiplicative component (2c) is violated. Those cases need to be treated differently.





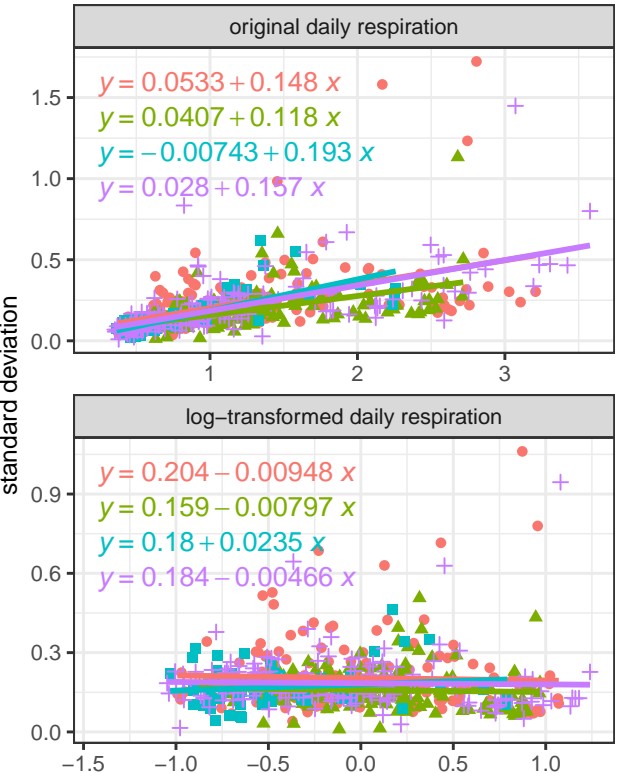

**Figure 2.** At original scale the error magnitude (standard deviation of error terms across day) scales with flux magnitude (top). Log-transformation avoids this problem (bottom). Different symbols and colors correspond to different chambers.

One way of counteracting this overestimation of uncertainty is setting a gapfilling flag for the uncertainty estimate of very small fluxes (Fig. 4, center penal). The remaining instances of wider confidence bounds with the lognormal assumption are due to lower number of original measurements in the uncertainty aggregation.

The few cases with narrower confidence intervals given the lognormal assumption occurred on days with limited original measurements and large outliers in estimated uncertainty of the single measurements. The lognormal approach was much less sensitive to large outliers and yielded narrower confidence intervals of the aggregated value. When constraining the dataset to days with at least 10 original measurements, then most of the differences disappeared (Fig. 4, bottom panel).

Contrary to the short term aggregation, distribution of annually aggregated fluxes of each chamber did not differ between the two approaches (Fig. 5). The skewness in the distribution of uncertainty of annual estimates almost disappeared, as seen by the similar distance to upper and lower confidence bounds in Fig. 5.

### 3.4 Annual plot-level fluxes

The combined temporal annual and spatial aggregation to the plot level can be done by two alternatives. With one alternative, temporal aggregation (using either normal or lognormal assumption) is done first, and spatial aggregation (using lognormal



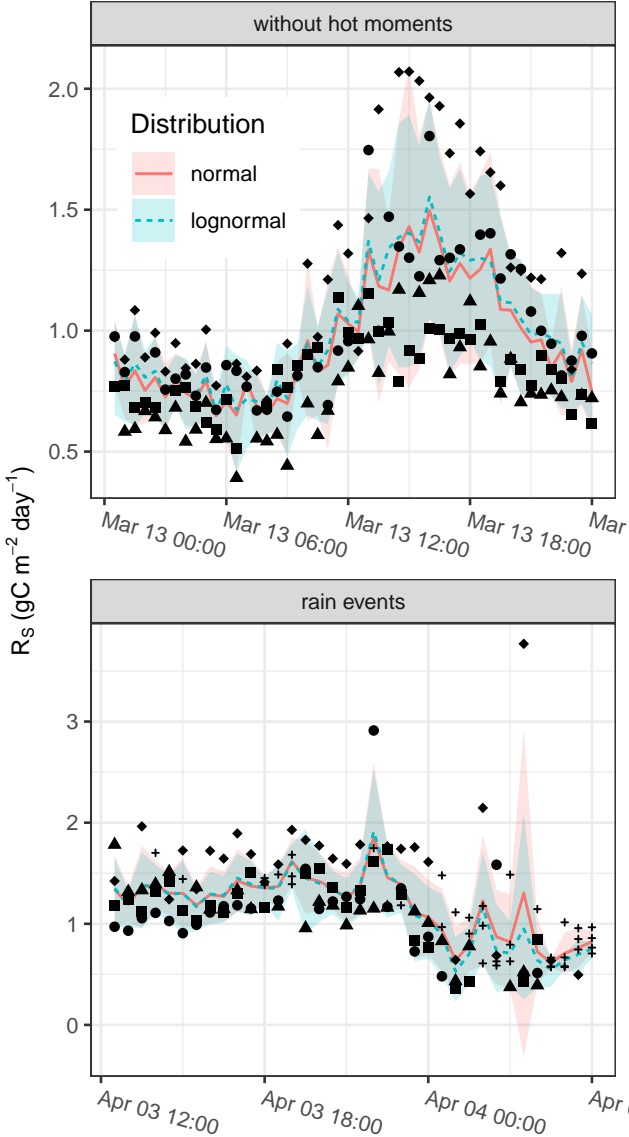

**Figure 3.** Observed fluxes for neighboring chambers (symbols), and spatially aggregated estimates: expected values (lines) and 95% confidence bounds (shaded areas). Crosses denote gapfilled values. The lognormal approach avoided negative lower confidence bounds with hot moments, e.g. with rain events on April 4th.

assumption) of single chamber annual flux estimates is done last. With the second alternative, the spatial aggregation is done first for each half-hour and these plot-level fluxes are then aggregated across time. The "space first" alternative yielded lower uncertainty estimates because it wrongly assumed true replicates when in reality there are only pseudo-spatial replicates (Fig. 6), i.e. locations did not change between successive measurements.





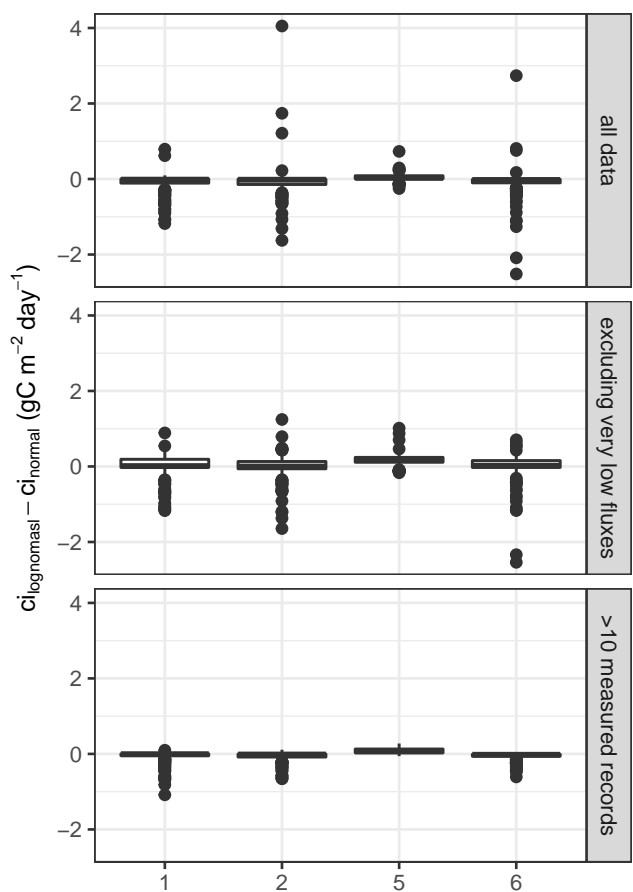

**Figure 4.** The difference in width of the 95% confidence intervals for daily aggregates between distributional assumption are shown by boxplots for four separate chambers (x-axis). Confidence ranges differed only for few cases with very low fluxes or with few original measurements.

## 4 Discussion

### 4.1 Improvement on distributional problems

With the lognormal assumption the distribution of random error can be inspected at log scale rather than at original scale. This improves two distributional problems (Savage et al., 2008). First, the lognormal distribution better approximates the more frequent occurrence of large errors (Fig. 1). And, second, the heteroscedastic nature of the random error is reduced, i.e. at log scale residual variance does not increase with flux magnitude (Fig. 2).

The increase of variance with flux magnitude also created the pattern of apparent Laplace distribution (Fig. 1). When we inspected the distribution of subsets of flux errors with similar magnitudes (using LUT section 2.3), we did not find the Laplace shape. This finding suggests that it is the superposition of normal distributions with different variance at different flux



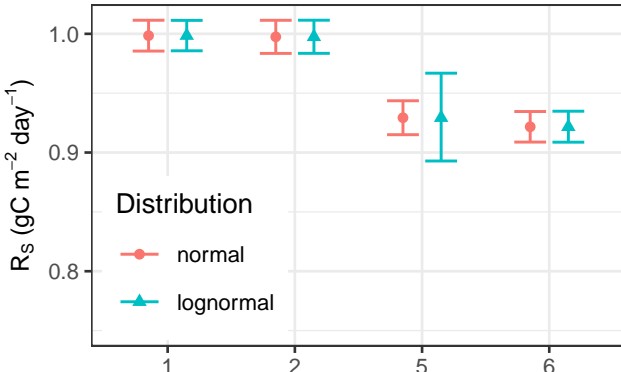

**Figure 5.** Annually aggregated flux estimates (symbols) and their confidence bounds (bars) are similar with normal and lognormal assumption. x-axis denotes different chamber locations.

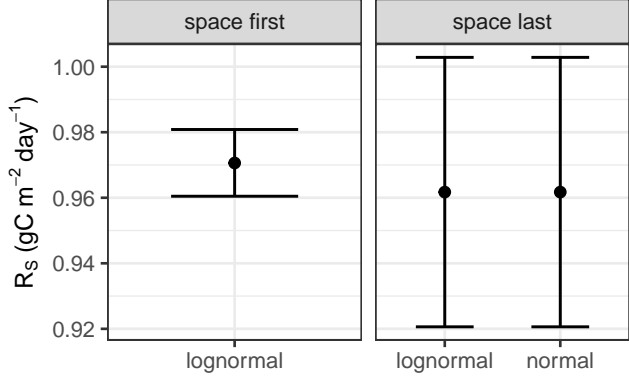

**Figure 6.** Annual plot-level flux estimates (symbols) had narrower 95% uncertainty bounds (bars) when wrongly aggregating over a time series of plot-level records (space first), compared to spatially aggregating annual fluxes of single chambers from figure 5 (space last).

magnitudes that leads to the apparent Laplace shape. This finding is similar to what Lasslop et al. (2008) found for random error of NEE that was measured by eddy covariance. Hence, when the error magnitude is used in model data integration exercises, we argue against using the Laplace assumption and against the associated usage of median absolute deviations (Richardson et al., 2006) when model predictions are compared to single observations.

5  **4.2  Spatial aggregation**

We argued that several spatially distributed chambers correspond to samples of a lognormal distribution. Using the lognormal assumption was especially important for periods of high spatial variability, i.e. differences between different chambers, which occurred at the Majadas de Tietar site mostly during dry summer period, similar to findings of Leon et al. (2014). Without using the lognormal assumption, confidence bounds of plot-level fluxes would include negative values (Fig. 3).





### 4.3 Daily temporal aggregation

Further, we explored consequences of aggregating measurements of a single chamber across time using the lognormal assumption compared to classical aggregation using the normal assumption. A single chamber measurement representing a time period can be assumed to be a normal or a lognormal random variable. These assumptions resulted in different aggregated un-

certainties when aggregating across a few days (Fig. 4). We argue that the choice of distributional assumptions depends on the sampling interval, the magnitude of measurement error, and the autocorrelation length of the process error. If measurements are frequent relative to process autocorrelation length, the uncertainty is dominated by the instrumentation error (from the measurement device), which can be assumed to follow a normal distribution. Alternatively, if a single measurement represents a longer period, the uncertainty will be dominated by process error. While process error dominates random error at a daily

measurement resolution (Lavoie et al., 2015), we cannot distinguish between those two cases from our series of half-hourly measurements.

However, we encountered a problem when fluxes were very low, where the instrumentation error component becomes dominant and the lognormal assumption is violated. If the lognormal assumption is applied to such cases, time aggregation leads to overestimation of uncertainty, because it overestimates the multiplicative error (Fig. 4). Those records need to be

filtered before aggregation using the lognormal approach.

### 4.4 Annual temporal aggregation

When aggregating half-hourly measurements of a single chamber to longer time scales such as to annual aggregates, the differences between distributional assumptions almost disappeared (Fig. 5). This was a consequence of relative uncertainty decreasing with the number of aggregated measurements which leads to less skew and to lognormal distributions which are

close to normal (Fig. A1)

Overall, we suggest using the lognormal assumption for a sample of spatially distributed fluxes, but the normal assumption for aggregating half-hourly observations of single chambers across time.

When deciding whether to first aggregate across space or time, the "space first" alternative (Fig. 6) wrongly assumes that the spatially aggregated values are only correlated in time. They are however, measured at the same spatial locations, and

fully correlated in space. Therefore, whenever measurement locations are fixed and a plot-level estimate is required, the spatial aggregation should be computed as the last step.

### 4.5 Process error

Our finding on the suitability of the lognormal assumption for random error sheds new light on the process error, i.e. the yet unattributed soil processes that generate random fluctuations in soil $CO_2$ efflux observations. Lavoie et al. (2015) proposed two

mechanisms for process error. First, a higher diversity of active metabolic pathways associated with a wider range of pore scale respiration rates at high temperature could result in larger variability of fluxes. Because higher temperatures are associated to higher fluxes, this would explain the increase of variance with flux magnitude. Second, gas diffusion rates might increase due





to heat produced during respiration. Similarly, gas transport processes in soil can change with pore space varying with soil moisture (Maier et al., 2011).

We propose the alternative hypothesis of stochasticity due to microbial dynamics and small scale spatial heterogeneity. Microbial communities differ across micrometer distances and vary in their sensitivity to temperature. Respiration is related to temperature sensitivity in an exponential manner (Lloyd and Taylor, 1994). Hence, if variation in temperature sensitivity is normally distributed, then the log of respiration is normally distributed, i.e., variation in respiration is log-normally distributed. This argument is transferable to process error distribution of respiration at the leaf and ecosystem scale.

Another mechanism that could result in lognormal process error are microbial self-enforcing feedback loops. The soil $CO_2$ efflux is a sum of the respiration of disconnected microbial communities at soil pore scale. Many of the communities show apparent stochastic dynamics. For example, for a given substrate concentration an already large community producing a suffi­cient enzyme concentration can thrive, while for a small community the enzyme concentration is to low for depolymerization to provide sustainable growth (Vasilyeva et al., 2016; Kaiser et al., 2017; Wutzler et al., 2017). Whether a microbial colony is in dormancy or exponential growth is determined by the very fine details of its history and environment at the soil pore scale. For given environmental conditions, we can stochastically observe large or small fluxes depending on how many and how strongly communities are in exponential growth. The exponential growth of microbes and self-enforcing positive feedbacks are in line with the multiplicative nature of observed soil respiration and occurrence of large positive extremes.

### 4.6 Recommendation checklist

To obtain plot-level estimates of soil $CO_2$ efflux one typically has to aggregate time series of several chambers. For such cases we recommend the following procedure based on the experience gained with this study.

- Estimate error terms by daily differencing or, preferentially, LUT.

- Fill gaps in the data and indicate which records are gap-filled.

- Aggregate data of single chambers across time and take care of autocorrelation. Use the normal assumption for high-frequency, e.g. hourly measurements, or use the lognormal assumption if a single measurement represents a period of several hours. For the latter case, flag the low flux conditions, where instrumentation error becomes dominant.

- At the last step, aggregate the time-aggregated estimates across several chambers using the log-normal assumption.

- In model-data integration compare predictions and observations of soil $CO_2$ efflux at the log-scale.

## 5 Conclusions

The presented methodology and tools[2] will help researchers to better analyse soil $CO_2$ efflux measurements using different assumptions.

---

[2]https://github.com/bgctw/lognorm





The lognormal assumption improves two error distribution problems, first, the strong tails, and second, the heteroscedasticity, i.e. the increase of error terms variance with flux magnitude. Hence, model data integration studies should consider comparing model predictions and observations at log-transformed scale.

The lognormal assumption is arguably better than the normal assumption when aggregating over spatially replicated cham-
bers. Researchers are encouraged to compute and report the parameters of the lognormal distribution. For aggregation of high-frequency flux measurements of a single chamber the normal assumption is plausible.

Whenever plot-level estimates are required, then spatial aggregation should be performed as the last step after temporal aggregation in order to avoid the fallacy of pseudo-replicates.

The lognormal assumption provides a new perspective on the yet unattributed processes responsible for process error in
fluxes. It implies that these processes operate in a multiplicative rather than in an additive way. The presented argument of respiration being exponentially related to a fluctuating temperature sensitivity is also true for leaf and ecosystem respiration. Hence we suggest testing whether or not the variability of such fluxes is better described by a lognormal distribution.

*Code and data availability.*   The essential functions to deal with the lognormally distributed measurements and their aggregation have been implemented in the openly available R package `lognorm`[2](doi will be generated for the version at final submission)
The code and the data generating the results and figures of this study are available upon request to the main author. We intend to publish the data used here as part of a bigger MaNiP project dataset.

*Author contributions.*   Thomas Wutzler analyzed the data and took the lead in writing the manuscript. All authors contributed to the writing and discussion. Kendalynn Morris and Tarek El-Madany maintained the chambers. Mirco Migliavacca designed the MaNiP experiment and leads the group.

*Competing interests.*   No competing interests are present

*Acknowledgements.*   Tarek El-Madany, Mirco Migliavacca, Oscar Perez-Priego, and Kendalynn Morris thank the Alexander von Humbold Stiftung for financial support of the MaNiP project. We want to thank Marco Pöhlmann, Olaf Kolle, Martin Hertel, Gerardo Marcos Moreno, Ramón López-Jimenez and Arnaud Carrara for helping us to maintain the automatic respiration chambers.





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

## Appendix A: The lognormal distribution

This section compiles the properties of the lognormal distribution that are most relevant in using the lognormal assumption in aggregating observations.

The density of the lognormal distribution is described by two parameters (A1).

$$f(x) = \frac{1}{x} \cdot \frac{1}{\sigma\sqrt{2\pi}} \exp\left(-\frac{(\ln x - \mu)^2}{2\sigma^2}\right) \tag{A1}$$

Traditionally, parameters are given at the log scale, where the location parameter $\mu$ describes the magnitude of a random variable and parameter $\sigma$ describes the spread. Their exponentials $\mu^* = e^\mu$ and $\sigma^* = e^\sigma$ describe the distribution at original scale with $\mu^*$ corresponding to the median and $\sigma^*$ being the the multiplicative standard deviation. The interval $\left(\mu^*/\sigma^{*2}, \mu^*\sigma^{*2}\right)$

denoted by $\mu^* {}^{\times}\!/ \sigma^{*2}$ contains about 95.5% of the probability mass.

The first two moments, i.e the expected value and the variance are given by (A2). The expected value is larger than the median, $\mu^*$ because the distribution is skewed to the left. With decreasing $\sigma^*$ the skewness decreases and the shape of the distribution gets closer to normal (Fig. A1).




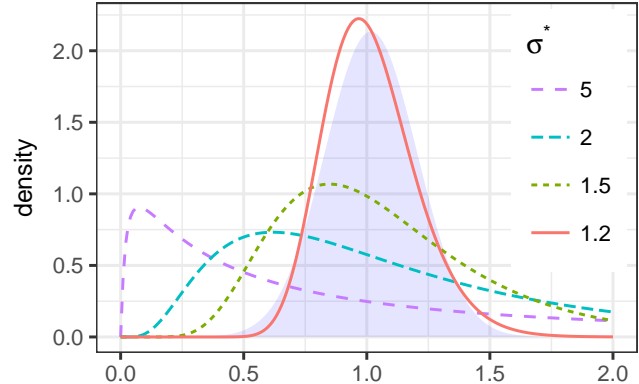

**Figure A1.** Density distributions of lognormal distributions (lines) get closer to normal density (shaded area) as multiplicative standard deviation $\sigma^*$ decreases down to 1.2 for same $\mu^* = 1$.

$$\mathrm{E}[X] = \bar{x} = \exp\left(\mu + \frac{\sigma^2}{2}\right) = \mu^* e^{\sigma^2/2} \tag{A2a}$$

$$\mathrm{Var}(X) = \sigma_o^2 = [\exp(\sigma^2) - 1]\exp(2\mu + \sigma^2)$$
$$= (e^{(\sigma^2)} - 1)\bar{x}^2 \tag{A2b}$$

Equation A2b relates the standard deviation to the relative error, i.e. the coefficient of variation: $\mathrm{cv} = \sigma_o/\bar{x}$ (A3). A relative
5   error of 5% corresponds to $\sigma^* = 1.05$ and approximating the lognormal distribution by a normal distribution worked reasonably well up to $\sigma^* = 1.2$ (Fig. A1) corresponding to a relative error of 18%.

$$\mathrm{cv} = \sqrt{e^{(\sigma^2)} - 1} \tag{A3a}$$

$$\sigma = \sqrt{\ln(\mathrm{cv}^2 + 1)} \tag{A3b}$$

The parameters of the distribution can be estimated by the log-transformed sample (A4).

10  $$\hat{\mu} = \mathrm{mean}\,(\ln x) \tag{A4a}$$

$$\hat{\sigma} = \mathrm{sd}\,(\ln x), \tag{A4b}$$

where $\mathrm{mean}(x) = \bar{x}$ and $\mathrm{sd}(x)$ denote the sample mean and standard deviation respectively. Alternatively, the distribution parameters can be also estimated from the mean and standard deviation at original scale, $\sigma_o$, by (A5) (Limpert et al., 2001).





$$\hat{\mu} = log\left(\bar{x}/sqrt(\omega)\right) \tag{A5a}$$

$$\hat{\sigma} = \sqrt{\ln(\omega)}, \tag{A5b}$$

where $\tag{A5c}$

$$\omega = 1 + \mathrm{cv}^2 \text{ and} \tag{A5d}$$

$\mathrm{cv} = \sigma_o/\bar{x}$ is the coefficient of variation. $\tag{A5e}$

The quantiles of the lognormal distribution are derived from the quantiles of the normal distribution (A6).

$$q_{\text{lognormal}}(p; \mu, \sigma) = e^{q_{\text{normal}}(p; \mu, \sigma)} \tag{A6}$$

For example, the 97.5% quantile of the standard normal distribution with , $q_{\text{normal}}(97.5, 0, 1) \approx 1.96$ directly translates to the lognormal, $q_{\text{lognormal}}(97.5, 0, 1) \approx e^{1.96}$. Hence, a 95% confidence interval of the normal is within $\mu \pm 1.96\sigma$ and that of

the lognormal is within $e^{\mu \pm 1.96\sigma}$, also denoted $\mu^* {}^\times\!/1.96\sigma^*$. Note that this confidence interval is not symmetrical with the upper bound being further away from the median.

The product of several lognormal random variables is again lognormally distributed, because the sum of normally distributed random variables at the log scale is again normally distributed.

For the sum of several lognormal random variables, to date, there is no closed formula known. However, it can be approx-

imated by a lognormal distribution and the parameters of this distribution can be found by various methods (Fenton, 1960; Cobb et al., 2012; Lo, 2013; Messica and Messica, 2016). In this study we use the approximation by Lo (2013), which can be applied to the sum of correlated random variables (A7).

$$S_+ = \mathrm{E}\left[\sum_i X_i\right] = \sum_i \mathrm{E}[X_i] = \sum_i e^{\mu_i + \sigma_i^2/2} \tag{A7a}$$

$$\sigma_S = 1/S_+^2 \sum_{i,j} \rho_{ij} \sigma_i \sigma_j e^{\mu_i} e^{\mu_j} \tag{A7b}$$

$\mu_S = \ln(S_+) - \sigma_S^2/2$ $\tag{A7c}$

where $S_+$ is the expected value of the sum, i.e the sum of the expected values of the terms. $\mu_s$ and $\sigma_S$ are lognormal distribution parameters of the sum, $\mu_i$ and $\sigma_i$ are the lognormal distribution parameters of the added random variables, and $\rho_{ij}$ is the correlation between two added random variables at log scale.

The multiplicative standard deviation, $\sigma^*$, is invariant to multiplications of the random variable. Hence, its the same for

the mean as for the sum of several lognormally distributed random variables. For the mean only the scale parameter changes compared to the sum as $\mu_{\text{mean}}^* = \mu_S^*/n$ or $\mu_{\text{mean}} = \mu_S - \ln(n)$, where $n$ is the number of aggregated variables.