# Peer review of "Soil $CO_2$ efflux errors are lognormally distributed - Implications and guidance."

_Geoscientific Instrumentation, Methods and Data Systems, 2019_

## Referee Comment (RC1) · Anonymous Referee #1 · 16 Sep 2019

**The application of the log-normal approach can be made even more convincing and consistent**

**Contents:**

Random errors are usually modelled with a normal distribution and a common error standard deviation. The paper shows that both assumptions are inadequate, at least for single measurements from soil CO2 efflux devices. These findings may well generalize to many other environmental measurements. The alternative model of a log-normal, multiplicative random error appears much more suitable and plausible and leads to more efficient and appropriate analyses.

**Theory:**

The theory on the lognormal distribution, the distribution of sums of such variables and the variance of sums of correlated variables is nicely summarized.

There is, however, a misnomer: The notion of "confidence interval" should be reserved for the interval that covers the true PARAMETER value with the given confidence level, but it is used to name an interval in which approximately 95% of the observations should be contained. This may be called a scatter interval. It is related, but not identical to a prediction interval or a tolerance interval.

**Application:**

Fig. 1: The comparison of the two main methods - using the normal or the lognormal assumption - is first examined by a respective qq plot of residuals. The residuals are obtained as the differences between 4 individual observations and their average either on the original or the log scale. However, these 4 observations stem from 4 measurement devices in 4 fixed places, and an inspection of Fig. 3 shows that they are clearly subject to systematic differences. Thus, the 4 residuals do not represent 4 independent random errors. In addition, the 4 residuals are collected from many half hours and then shown in a single qq plot. As the authors show in Fig.2 (and also emphasize in the text), these groups of 4 do not have the same variances in case of the "normal method". Therefore, they should not be shown in a common qq plot.

Fig. 2 (left panel) and the related comments show quite convincingly that the assumption of constant standard deviations does not hold - they rather increase with expected values. This disappears when the lognormal distribution is used (right panel), since for this model, the standard deviation is proportional to the expectation. However, an alternative would be to model the random error as normal with standard deviation proportional to the mean. (Note that I would NOT prefer this approach over the log-normal model!)

Fig. 4 shows that the scatter interval ("confidence interval") for values aggretated over a day are most often considerably narrower for the lognormal method. In the text, the

authors call them "the same" and discuss the few exceptions instead. Note, however, that one should first make sure that the intervals produced by the two methods indeed show approximately the same percentage of covering the observations.

In summary, while the theory has a good potential to improve the methodology, the way it is applied is not convincing.

**A suggestion:**

Fig. 3 suggests that the measurements follow a model

$$Y_{tk} = h(t) \cdot \gamma_k \cdot \epsilon_{tk}$$

or, on the log scale,

$$\log(Y_{tk}) = g(t) + \beta_k + \log(\epsilon_{tk})$$

where $t$ is the time, $k$, the measurement device (chamber), $g() = \log(h())$, smooth functions of time, and $Y_{tk}$ and $\epsilon_{tk}$ are the observations and the (lognormal) random error. Thus, it would be adequate to fit this model (on the log scale) and then show its adequacy using diagnostic plots (residuals against fitted values and time, qq plot).

(The smooth function g may - at least for other target variables than CO2 efflux - advantageously be related to explanatory variables such as a daily and/or a seasonal cycle and environmental variables, still allowing for a smooth additional term.)

This model can be fitted to half hourly measurements or daily averages. (The averages may be the usual arithmetic means or robust version of them.)

If the above model fits well, without showing heteroscadasticity, an alternative method for aggregating measurements to daily averages may be used: the fitted values can be aggregated, and the correction factor for getting an expectation from the first parameter of a log-normal, $\exp(\sigma^2/2)$, can be applied to the result. This is similar in spirit as using the estimated parameters of the lognormal distribution obtained from the 4 measurement devices, as done in the paper, but that method wrongly includes the systematic

effect $\gamma_k$ in the random variability of the error term.

**Discussion:**

The model just described has consequences for

- spatial aggregation: The dominant term in such means are the mean of the $\gamma_k$. They can be interpreted as random effects if the locations of the measurement devices are randomly chosen from the plot (or region). This is why aggregating over time first ("space last") produced a more plausible interval (Fig. 6). The authors correctly explain this fact in other words - that this occurs "because it wrongly assumed true replicates when in reality there are only pseudo-spatial replicates."

- temporal aggregation: If there is a daily or seasonal cycle, the time points should not be treated as a stationary time series. If the smooth function $g$ looks like a stationary function without further patterns, then it can be integrated into the correlated error term.

The authors say: "We argued that several spatially distributed chambers correspond to samples of a lognormal distribution."
It would be difficult to judge the distributional form from 4 independent values $\gamma_k$. Thus, this amounts to an over-interpretation.

**Process error:**

The authors give many reasons why the magnitude of the process error may depend on the magnitude of the CO2 efflux itself. In fact, in biology and other fields, all kinds of variation typically depend on the magnitude of the variable under consideration, and it would be more necessary to find explanations if for some process this is not the case.

This is even true for measurement errors. I do not see an intrinsic reason why the measurement error should be better described by a normal than a log-normal distribution.

Hopefully, it is small enough that the two models cannot be distinguished. The authors might adapt their comments on measurement errors accordingly.

This contradicts their recommendation to "use the normal assumption for high- frequency, e.g. hourly measurements," for which I do not find a convincing reason.

**Recommendations:**

These might be recast along the following line:
The lognormal model and methods are more appropriate for this kind of data. The feature of standard deviations being proportional to expectations is the dominant tigger of the improvements. When measurements are aggregated to means or sums, the central limit theorem grants that the fit of the normal distribution gets better, but it will rarely be better than the fit of the log-normal for environmental and similar measurements, even after aggregation. However, the use of the log-normal is most needed when considering single measurements, for example, when they are compared with model values from any models (as the authors mention it).

Since this discussion suggests major changes of the analyses and interpretations, I do not go into details of the text at this stage.

---

## Author Comment (AC1) · 2 Oct 2019

We thank anonymous Referee #1 (RC1) for his encouraging and constructive comments.

Find the full comment in the attached pdf. Here, we summarize the main points.

RC1 suggest to make the lognormal assumption the default case, sees a need to argue for normal assumption instead, and wishes to change the interpretation accordingly. In our discussion, we tried to be conservative and accepted the alternative assumption of lognormal distribution only with those scenarios that showed a difference and improvement compared to the business-as-usual of the normal/Gaussian assumption. In regard of the intended audience, we hesitate to adopt this stronger point of view and

would rather stick to the more conservative default of the normal assumption.

RC1 also suggested a revised model for analysis and fitting all chambers together with this model and hence change the analysis. We agree that such an analysis was more elegant and concise. However, we applied the analysis and see several discrepancies of the suggested model (possible new appendix C). The main problem is that chamber effects and properties of the error structure vary with time. While there might be statistical models that can deal with these issues, itwill be difficult for many readers to replicate with their tool-set. Therefore, we wand to stick to the current method based on shifting time windows. In this journal that primarily deals with measurements and their processing, we plan to keep the current approach of estimating true values by the LUT algorithm and focus on the structure of the resulting residuals. The LUT approach is familiar to ecologists working on Eddy-Covariance sites.

We want to add an additional appendix B, where we extend our work with fitting both error terms (instrumentation error at normal scale and process error at log scale) using bayesian hierarchical modelling. Although the necessary statistical tools and their explanation go beyond the basic message and guidance of this paper, we are interested in the opinion of RC1.

Please also note the supplement to this comment:
https://www.geosci-instrum-method-data-syst-discuss.net/gi-2019-10/gi-2019-10-AC1-supplement.pdf

**Supplement:**

We thank anonymous Referee #1 (RC1) for his encouraging and constructive comments.

In the first part of this reply we respond on the general issues and in the second part we answer to the specific statements of RC1.

RC1 suggest to make the lognormal assumption the default case, sees a need to argue for normal assumption instead, and wishes to change the interpretation accordingly.
In our discussion, we tried to be conservative and accepted the alternative assumption of lognormal distribution only with those scenarios that showed a difference and improvement compared to the business-as-usual of the normal/Gaussian assumption. In regard of the intended audience, we hesitate to adopt this stronger point of view and would rather stick to the more conservative default of the normal assumption.

RC1 also suggested a revised model for analysis and fitting all chambers together with this model and hence change the analysis. We agree that such an analysis was more elegant and concise. However, we applied the analysis and see several discrepancies of the suggested model (possible new appendix C). The main problem is that chamber effects and properties of the error structure vary with time. While there might be statistical models that can deal with these issues, itwill be difficult for many readers to replicate with their tool-set.
Therefore, we wand to stick to the current method based on shifting time windows. In this journal that primarily deals with measurements and their processing, we plan to keep the current approach of estimating true values by the LUT algorithm and focus on the structure of the resulting residuals. The LUT approach is familiar to ecologists working on Eddy-Covariance sites.

We want to add an additional appendix B, where we extend our work with fitting both error terms (instrumentation error at normal scale and process error at log scale) using bayesian hierarchical modelling. Although the necessary statistical tools and their explanation go beyond the basic message and guidance of this paper, we are interested in the opinion of RC1.

In the following we reply to the specific points. Quotes of of RC1 are preceded by RC1-<counter>: and noted in blue color. The counter is intended to help referring to specific points.

RC1-1: The theory on the lognormal distribution, the distribution of sums of such variables and the variance of sums of correlated variables is nicely summarized.
There is, however, a misnomer: The notion of "confidence interval" should be reserved for the interval that covers the true PARAMETER value with the given confidence level, but it is used to name an interval in which approximately 95% of the observations should be contained. This may be called a scatter interval. It is related, but not identical to a prediction interval or a tolerance interval.

Thanks for noting this misnomer. In Figs 3 and 4 we fit parameters of alternative distributions to a sample of observations and report intervals of quantiles of the fitted distribution. This corresponds to the interval in which a single future observation ( "prediction interval") or multiple future observations ("tolerance interval") of this distribution will fall. We will change notation from "confidence interval" and "confidence bounds" to "prediction interval". Although we acknowledge the "scatter interval" or "tolerance interval" maybe technically better fitting terms, we anticipate that the intended audience (including ourselves) would be irritated by those unfamiliar terms, whose subtleties are not central to the message of this paper. A quick web search did not lead to any explanation of "scatter interval".

Could you, please, guide us to relevant sources of more in depth explanation of the differences between these terms?

RC1-2: Fig. 1: The comparison of the two main methods - using the normal or the lognormal assumption - is first examined by a respective qq plot of residuals. The residuals are obtained as the differences between 4 individual observations and their average either on the original or the log scale. However, these 4 observations stem from 4 measurement devices in 4 fixed places, and an inspection of Fig. 3 shows that they are clearly subject to systematic differences. Thus, the 4 residuals do not represent 4 independent random errors. In addition, the 4 residuals are collected from many half hours and then shown in a single qq plot. As the authors show in Fig.2 (and also emphasize in the text), these groups of 4 do not have the same variances in case of the "normal method". Therefore, they should not be shown in a common qq plot.

Correct. Therefore, we had produced these statistics and plots for each chamber separately. Fig 1 shows the graphs for chamber 2 (the one with the least non-missing records). The plots for the other chambers to not look very different. We will explain this better in the Figure caption.

RC1-3: Fig. 2 (left panel) and the related comments show quite convincingly that the assumption of constant standard deviations does not hold - they rather increase with expected values. This disappears when the lognormal distribution is used (right panel), since for this model, the standard deviation is proportional to the expectation.

This is exactly what Fig 2 should communicate.

RC1-4: However, an alternative would be to model the random error as normal with standard deviation proportional to the mean. (Note that I would NOT prefer this approach over the log-normal model!)

We will refer to the mentioned but not preferred alternative in the text.

RC1-5: Fig. 4 shows that the scatter interval ("confidence interval") for values aggretated over a day are most often considerably narrower for the lognormal method. In the text, the authors call them "the same" and discuss the few exceptions instead. Note, however, that one should first make sure that the intervals produced by the two methods indeed show approximately the same percentage of covering the observations. In summary, while the theory has a good potential to improve the methodology, the way it is applied is not convincing.

To our interpretation, the box plots of Fig 4 show only for very few cases (denoted by outlier dots) a narrower prediction interval for the lognormal assumption. The mass (denoted by very flat boxes) centers at a difference in interval width of zero. The few outlier dots are identified by +/-1.58 IQR/sqrt(n), which roughly corresponds to being outside the 95% confidence interval (state in its help-page).

RC1-6: A suggestion:
Fig. 3 suggests that the measurements follow a model
Y tk = h(t) · γ_k · eps_tk

$\log(Y_{tk}) = g(t) + \beta_k + \log(\#_{tk})$
where t is the time, k, the measurement device (chamber), $g() = \log(h())$, smooth
functions of time, and $Y_{tk}$ and $\#_{tk}$ are the observations and the (lognormal) random
error. Thus, it would be adequate to fit this model (on the log scale) and then show its
adequacy using diagnostic plots (residuals against fitted values and time, qq plot).
(The smooth function g may - at least for other target variables than $CO_2$ efflux - advan-
tageously be related to explanatory variables such as a daily and/or a seasonal cycle
and environmental variables, still allowing for a smooth additional term.)
This model can be fitted to half hourly measurements or daily averages. (The averages
may be the usual arithmetic means or robust version of them.)
If the above model fits well, without showing heteroscadasticity, an alternative method
for aggregating measurements to daily averages may be used: the fitted values can be
aggregated, and the correction factor for getting an expectation from the first parameter
of a log-normal, $\exp(\sigma^2/2)$, can be applied to the result. This is similar in spirit as using
the estimated parameters of the lognormal distribution obtained from the 4 measure-
ment devices, as done in the paper, but that method wrongly includes the systematic

Thank you for proposing this new model. We studied in with suggested appendix C but with a slightly
different interpretation.
First, the observation error eps_tk is additive (see reply to RC1-11). The suggested model considers a
multiplicative observation error. Second, the process error is probably related but not the same as the
differences between chambers. It is present also in the time series of a single chamber.  Hence it cannot
be captured by the index k in a random effect offset across the chambers.

Hence, we reinterpret this model as an approximation where observation error is neglected, eps_tk is
the single chamber process error, and  γ_k is the cross-chamber process error.

However, the model does not fit well. The random effect of the chamber is also a function of time.
Although it changes much more slowly (correlation length of a several weeks) than h(t), which has a
daily pattern. We will work on representing the chamber offsets as Gaussian processes but this requires
us to learn more about hierarchical bayesian models and this will be beyond the scope of this paper.

RC1-7: Discussion:
The model just described has consequences for
• spatial aggregation: The dominant term in such means are the mean of the γ k .
They can be interpreted as random effects if the locations of the measurement
devices are randomly chosen from the plot (or region). This is why aggregating
over time first ("space last") produced a more plausible interval (Fig. 6). The
authors correctly explain this fact in other words - that this occurs "because it
wrongly assumed true replicates when in reality there are only pseudo-spatial
replicates."

We fully agree, but see our discussion on the temporal patterns of y_k and our concerns of fitting all
chambers with one model.

RC1-8: • temporal aggregation: If there is a daily or seasonal cycle, the time points should
not be treated as a stationary time series. If the smooth function g looks like
a stationary function without further patterns, then it can be integrated into the

We agree and already argued that it is the non-stationarity of the chamber effects that is the most pressing problem. This is why we prefer the time-window based LUT estimate compared to fitting the full model.

RC1-9: The authors say: "We argued that several spatially distributed chambers correspond to samples of a lognormal distribution."
It would be difficult to judge the distributional form from 4 independent values $\gamma k$. Thus, this amounts to an over-interpretation.

We agree that this would be over-interpretation and we cannot infer this from our 4 spatial replicates. We extrapolated our insight of comparing the consequences of the normal vs. the lognormal assumption of errors from single time series to several chambers.

In order to make this extrapolation more explicit, we suggest the formulation formulation:

"We argued that if a lognormally distributed process error dominates the observation error of single chamber, then such a processer error is assumed to dominate the differences between chambers. Hence, we assume also a lognormal distribution of measurements across several chambers."

RC1-10: Process error:
The authors give many reasons why the magnitude of the process error may depend on the magnitude of the CO2 efflux itself. In fact, in biology and other fields, all kinds of variation typically depend on the magnitude of the variable under consideration, and it would be more necessary to find explanations if for some process this is not the case.

We fully agree. Although for the purpose of this paper we argue from a more conservative view with a Gaussian default.

RC1-11: This is even true for measurement errors. I do not see an intrinsic reason why the measurement error should be better described by a normal than a log-normal distribution. Hopefully, it is small enough that the two models cannot be distinguished. The authors might adapt their comments on measurement errors accordingly.

In our view, the measurement error is primarily a result of the measurement device. Depending on the measurement device both cases are possible (increasing or not increasing with the magnitude of the flux). For flux chambers, there are many studies that look at the various sources of the systematic and random errors. Calibrations against known fluxes, here, do not show apparent deviations from Gaussian or scaling with flux magnitude. Moreover, especially on very small fluxes, our data shows considerable measurement errors and also negative fluxes. So we do now that in our case there is an error component that does not decrease (scale) with the flux magnitude.

RC1-12: This contradicts their recommendation to "use the normal assumption for high- frequency, e.g. hourly measurements," for which I do not find a convincing reason.

We see the point of the reviewer that if measurement error also scales with the flux, this will be true. However, we argued in RC 1-11 that in our case measurement error does not scale with flux magnitude.

For high-frequency fluxes, we argue that we cannot show that magnitude of the random error is larger than the additive measurement error and therefore, on conservative grounds, do not recommend a change of current processing approach. We so far intend to stick to this conservative perspective in the revised version.

RC1-13: Recommendations:
These might be recast along the following line:
The lognormal model and methods are more appropriate for this kind of data. The feature of standard deviations being proportional to expectations is the dominant tigger of the improvements. When measurements are aggregated to means or sums, the central limit theorem grants that the fit of the normal distribution gets better, but it will rarely be better than the fit of the log-normal for environmental and similar measurements, even after aggregation. However, the use of the log-normal is most needed when considering single measurements, for example, when they are compared with model values from any models (as the authors mention it).

We hesitate to write from such a strong point of log-normal as the default case in this paper and adopt the normal case as default and collect arguments where the log-normal should be applied instead. We want to convince practitioners to rethink their business-as-usual of the normal assumption and convince them to invest time to investigate the alternative usage of the log-normal assumptions. For this purpose we want to write from a more conservative perspective and argue for evidence for the lognormal assumption.

RC1-14: Since this discussion suggests major changes of the analyses and interpretations, I do not go into details of the text at this stage.

We look forward to a fruitful discussion on the proper model for the analysis of soil efflux data. We investigated the model proposed by RC1 (appendix C) and look forward to RC1 response to our clarified updated model of appendix B. This issue is relevant for many more data processing that we ecologists deal with.

**Appendix B: Latent Gaussian Model formulation**

The observations of CO2 efflux of a single chamber can be formulated as a Latent Gaussian Model (LGM) a subset of Bayesian hierarchical models (Rue et al., 2009).

$$R_S = R_\mu + \epsilon_{IE} \tag{B1a}$$

$$\ln(R_\mu) = R_B + \epsilon_{PR} \tag{B1b}$$

$$\epsilon_{IE} \sim N(0, \sigma_{IE}) \tag{B1c}$$

$$\epsilon_{PR} \sim N(0, \sigma) \tag{B1d}$$

$$\ln(\sigma_{IE}^{-2}) \sim \text{Gamma}(a_{IE}, b_{IE}) \tag{B1e}$$

$$\ln(\sigma^{-2)}) \sim \text{Gamma}(a, b) \tag{B1f}$$

where $\epsilon$ are normally distributed error terms with shape parameter $\sigma$. Their corresponding precisions $1/\sigma^2$ are distributed by a logGamma hyperprior with specified parameters. $R_b$ is the true value of log (soil $CO_2$ efflux). It can be either plugged in by the LUT approach, modelled as a linear model of covariates, or estimated together with the $sigma$ parameters given a proper constraint on their covariances in time and/or space of environmental variables.

Such a LGM can be estimated using INLA (Rue et al., 2009) (appendix B3) or Markov-Chain Monte-Carlo Sampling (Metropolis et al., 1953; Gelman et al., 1995; Zobitz et al., 2011).

Instrumentation error is of magnitude $IE = \sigma_{IE}$ and process error at original scale is of magnitude $PR = R_B(e^\sigma - 1)$. This study deals with two special cases. Since, the log-normally distributed process error scales with the flux magnitude, we expect it to dominate at large fluxes, while we expect the instrumentation error to dominate at low fluxes.

**B1  $PR << IE$**

If the log-normally distributed error is small compared to the normally distributed one, it can be neglected. The model then simplifies to eq. B2.

$$R_S = R_{Be} + \epsilon_{IE} \tag{B2a}$$

$$\epsilon_{IE} \sim N(0, \sigma_{IE}) \tag{B2b}$$

$$\ln(\sigma_{IE}^{-2}) \sim \text{Gamma}(a_{IE}, b_{IE}) \tag{B2c}$$

With $R_{Be} = e^{R_B}$. When assuming a flat prior ($a_{IE} = -1$, $b_{IE} = 0$), this model corresponds to a classical linear regression.

**B2  $IE << PR$**

If the normally distributed error is small compared to the log-normally distributed one, it can be neglected. The model then simplifies to eq. B3.

$$log(R_S) = R_B + \epsilon \tag{B3a}$$

$$\epsilon \sim N(0, \sigma) \tag{B3b}$$

$$\ln(\sigma_{IE}^{-2}) \sim \text{Gamma}(a, b) \tag{B3c}$$

When assuming a flat prior ($a = -1$, $b = 0$), this model corresponds to a classical linear regression of the log-transformed

5    values.

**B3    Fitting the LGM using INLA**

During periods with similar environmental conditions, we can model $R_B$ as a smooth function with time and fit it together
with the magnitudes of the two error types, $\sigma_{IE}$ and $\sigma$, without the need of gap-filling before.

We fitted model B1 to the data of chamber two for a 5-day period in April using INLA (Rue et al., 2009) and its default

10    priors and compared the posterior estimates of the two standard deviations. While the standard deviations were both significant
and of the same magnitude ($\sigma_{IE} : mean = 0.047, q025 = 0.021, q975 = 0.088$), the transformation of the lognormal error to
original scale indicated the deviations due to the lognormal error had a larger effect: $,R_B(e^\sigma - 1) : mean = 0.095, q025 = 0.024, q975 = 0.18$).

During shorter ($\sim$weeks) periods, we can assume that the difference between chambers can be modelled as a random inter-

15    cept+slope in the linear predictor at log scale. A simpler random intercept only model still showed patterns in the residuals.

Compared ot the single-chamber-fit, the estimate of the normal error decreased further ($\sigma_{IE} = 0.010$), whereas the estimate
of the lognormal error increased ($R_B(e^\sigma - 1) = 0.27$), and standard deviation of chambers intercept was larger: $R_B(e^{\sigma_{Ch}} - 1) : mean = 1.0, q025 = 0.55, q975 = 1.9$).

This indicates that assumption of negligible instrumentation error compared to the log-normally distributed process error

20    ($IE << PR$) is viable.

In addition, to the model with both error terms, we fitted models with only one of the error terms included and compared
models by DIC criterion (Spiegelhalter et al., 2002). The much lower DIC of full model of -5130 indicated a much better fit
compared to the lognormal-error-only model with DIC of -1080 and the normal-error-only model with a DIC of 555.

We will study if this approach can be extended to longer periods and adapted to more complex models of differences between

25    chambers.

**Appendix C:  Lognormal fixed effects model**

Using assumption $IE << PR$, we tried fitting a mixed effects model to the log-transformed observations of the entire non-
gapfilled dataset. First, we filtered for respiration being above $1.2 \, \text{gCm}^{-2}\text{day}^{-1}$ to exclude those records strongly influenced by
the instrumentation error. If residual plots showed adequacy of the model, an alternative method for aggregating measurements

30    to daily averages may be used.

**22**

[Figure]

**Figure C1.** Residuals of the GAMM fit to the entire log-transformed respiration data.

We used mgcv package in R with the following specification:

```
ml1 <- gamm(
R ~ temp + moist + s(time, bs = "cr") + chamber, data = dsFin
,weights = varPower(0.4, ~temp)
5 )
```

The model included fixed effects for soil temperature and soil moisture, a smoothing term for time and an offset per chamber. The increase of residual variance with flux magnitude or temperature was accounted for by the weights term.

However, when inspecting Pearson-Residual plots (figures C1, C2, C3), it became apparent, that this model was not adequate. Residuals showed a clear pattern with different chambers and a despite of the smoother varying patterns with time.

10   The most important problem was fluctuating pattern of chamber effects with time.

[Figure]

**Figure C2.** Subset of residuals of the GAMM fit to the entire log-transformed respiration data during five days in April.

[Figure]

**Figure C3.** Subset of residuals of the GAMM fit to the entire log-transformed respiration data during five days in October.

---

## Referee Comment (RC2) · Anonymous Referee #1 · 14 Oct 2019

General Model

Generally speaking, flux measurements (as any soil variables) can be seen as the result of a process and the effect of measurement. "Process error" is a suboptimal term and should be replaced by "process variation" – the part that is of interest in environmental research. It can be modeled in space and time. The time variation can be attributed partly to other environmental variables, and space variation is limited to the 4 chanbers in fixed locations in this application. Whatever is not adequately described by such models might then be split into a "process error" and the measurement error on the basis of particular assumptions about the measurement error.

The popular method LUT reflects this general idea. It assumes that the process part is a function of temperature and moisture, and therefore asks that the values to be

averaged must have respective values within a narrow range, and so should the hour of the day, which stands for other environmental conditions with a daily cycle. In addition, rain events are excluded.

Thus, the time variation can be modeled by – a regression on temperature, moisture – rain, with possible after-effects – possibly other explanatory variables – a daily cycle – a smooth function of time – a stationary fluctuation – the real "process error" – the location of the measurement device Such a model, if based on linear combinations (additive effects), is clearly more successful on a log scale. There is a measurement error that is additive in the non-logarithmic scale and may be assumed as normally distributed.

Describing such a general model, I do not suggest that implementing it should be the content of the paper discussed here. It just describes the background for my comments.

Replies to the replies

Reply1 "RC1 suggest to make the lognormal assumption the default case" and the authors give reasons why they "would rather stick to the more conservative default of the normal assumption" ... for the Instrument Error IE

The authors have argued in their reply that empirical evidence documented in the literature suggests that the variance of the IE does not depend on the flux. This is a convincing argument against my suggestion. It should be mentioned in the revised paper.

Reply2 "revised model": The suggestion has been taken up and documented as Appendix C in the extended Reply. The authors conclude: "However, we applied the analysis and see several discrepancies..."

I think that the model has shown a good potential to gain insight into the process. It intends to split what the authors treat as the "process error" PE (more adequately called

"process variation") into a (large) part that is explained by the model and a remainder that still needs to be modeled as a random deviation and might rightously be called "process error".

If fitted to short periods, it seems plausible that it fits well enough to act as a better alternative to the LUT method. The authors should decide if they want to include this second use into the paper. The full potential of the model certainly goes beyond the scope.

The authors also fit a Bayesian hierarchical model that allows for their combination of a lognormal "process error" and a normal IE. I do not think that modelling the process variation as an i.i.d. sample of a lognormal variable is adequate. Rather, such a combination can be added to the foregoing model to improve its adequacy and fit. It amounts to fitting a nonlinear regression with the model $Y = \exp(h) + e$ , where h is (the linear predictor of) the foregoing model.

Conclusion

In summary, the authors intend to implement some suggestions I have made, and they have convincingly argued against others. The resulting revised paper will be a very valuable contribution to the statistical methodology for the targetted field.

Please also note the supplement to this comment:
https://www.geosci-instrum-method-data-syst-discuss.net/gi-2019-10/gi-2019-10-RC2-supplement.pdf

---

## Referee Comment (RC3) · Anonymous Referee #2 · 25 Nov 2019

This manuscript uses a year of data from four continuous (30 minute) soil CO2 efflux chambers to examine how different distributional assumptions–in particular, whether one assumes that random errors are normally or lognormally distributed–affects inferred error structures, confidence intervals, and annual sums. This is an interesting topic and appropriate for GIMDS. The ms is reasonably well written, concise but insightful, and provides useful guidance and a convincing argument for researchers in this area.

There are a few problems. It would be useful to discuss negative fluxes in a bit more detail, because they do occur, and not just because of error (see below). Several of the figures should be rethought. Finally, no data and code availability is specified. The latter seems to me particularly important. . .ah, I see the footnotes at the bottom of p.

[Figure]

13. It would be good to feature this more prominently, as the authors 'lognorm' package might be of broad interest.

In summary, this is a short, technical, and very interesting look at the statistical assumptions researchers make when gap-filling and otherwise handling soil respiration data. It needs minor to moderate revisions.

Specific comments

=================

1. Bottom of p. 2 and top of p. 3: this is an interesting question. In fact, soil $CO_2$ efflux (the soil-to-atmosphere) flux can definitely be negative at times due to pressure effects for example, or soil drainage. The actual process of soil respiration can probably be negative in certain cases as well (for all these see e.g. https://www.sciencedirect.com/science/article/pii/S004896971531144X). Obviously *usually* soil respiration is a positive number, but I think a bit more nuanced discussion would be useful here

2. I share R1's concern about Figure 1, which seems to be pooled data from all four chambers. Is this appropriate? It seems better to show one line per chamber

3. It's difficult to see what's going on in Figure 2–things are jumbled and overplotted. Faceting by method (x) versus chamber (y) would separate the chambers' data individually and be clearer

4. Top of p. 8, "panel" not "penal"

5. Seems like Figure 6 would be more effective as a table, or removed entirely and simply reported in the text